# Survival probabilities of thornback skate (*Raja clavata*) and spotted skate (*Raja montagui*) discarded by tickler chain beam trawl, pulse trawl, and flyshoot fisheries

**Edward Schram** [1]*, **Lennert van de Pol**[1], **Katinka Bleeker**[1], **Pieke Molenaar**[1], **Allard van Mens**[1], **Jan Jaap Poos**[2], **Karolina Molla Gazi**[1], **Suzanne Cornelisse**[1], **Pim van Dalen**[1], **Wouter Suykerbuyk**[1], **Jurgen Batsleer**[1]

1 Wageningen Marine Research, Wageningen University & Research, IJmuiden, The Netherlands,
2 Aquaculture and Fisheries Group, Wageningen University & Research, Wageningen, The Netherlands

* Edward.schram@wur.nl

## Abstract

We measured discards survival probabilities of thornback (*Raja clavata)* and spotted skate (*Raja montagui*) in tickler chain beam trawling (5 trips, n = 183 for thornback skate, n = 137 for spotted skate), pulse beam trawling (9 trips, n = 94 for thornback skate) and flyshoot fishieries (4 trips, n = 137 for thornback skate, n = 24 for spotted skate). Survival probabilities were measured by captive observation for 15 to 25 days post catch. All fishery operations were conducted in the southern North Sea (ICES division 27.4.c) and in the Eastern English Channel (ICES division 27.7.d) according to the regular commercial practices of the fishing vessels. Trips were spread out over the seasons to account for the effect of variable environmental and fishing conditions on discards survival. Operational and environmental conditions during sea trips were recorded. For beam trawling survival probabilities (95% CI) were 50% (43–57) for thornback and 44% (37–54) for spotted skate. For pulse trawling, survival probability of thornback skate was 54% (40–65). For flyshooting survival probabilities were higher: 80% (73–87) for thornback and 75% (60–94) for spotted skate. Survival probabilities were significantly affected by gear, catch processing time, water temperature, wave height and the interaction between water temperature and wave height. We found no evidence for effects of species and length on survival probability. Vitality class provides a useful qualitative prediction of survival probability within gear. These findings are relevant for the fisheries management of skates stocks in the North Sea and English Channel.

## Introduction

Elasmobranchs (e.g. sharks, skates and rays) play important roles as predators in the marine ecosystem [1]. They are long-lived, slow growing, reach sexual maturity at relatively old age, and produce a small number of offspring per year. This intrinsic biological sensitivity [2] makes them vulnerable to overexploitation and changes in essential habitats, especially

**Data Availability Statement:** All relevant data are within the manuscript and its Supporting Information files.

**Funding:** European Union, European Maritime and Fisheries Fund (EMFF)

**Competing interests:** The authors have declared that no competing interests exist.

spawning and nursery areas [3–6]. Bycatch related overfishing threatens many elasmobranch species [7] and bycatches occur in fisheries ranging from small scale artisanal [e.g. 8] to industrial [e.g. 9]. Historically the North Sea was home to some 25 elasmobranch species including thornback (*Raja clavata*), spotted (*Raja montagui*) and blonde skate (*Raja brachyura*). The decline of skates (order: Rajiformes, Class: Chondrichthyes) in the North Sea, especially in the 20th century, is generally attributed to long-term intensive fishing as well as large-scale coastal infrastructure and pollution [6, 10, 11]. Despite the increasing number of threatened elasmobranch species in the Northeast Atlantic [2], fisheries independent data show an increase in abundance since 2010 for thornback skate (*Raja clavata*) and blonde skate (*Raja brachyura*) in the North Sea [12].

In the North Sea, skates are mainly caught as by-catch in mixed demersal fisheries for flatfish [13, 14]. Commercially the most important species are thornback, blonde, and spotted skate (*Raja montagui*). The exploitation of these species is regulated by a single, common Total Allowable Catch (TAC) [15], local management measures such as seasonal closures and gear restrictions, minimum landing sizes (MLS), and a landing obligation [16]. Landings are currently restricted to skates with total lengths above 45 to 55 cm, varying by country, and by weight-based landing limits per trip to prevent overexploitation. Skates below the MLS and surplus catches are discarded. Part of the discarded skates may not survive and information about their survival is important for modelling skate population dynamics for conservation and stock assessment purposes [17, 20] and for exemptions to the landing obligation. The landing obligation (LO), implemented under the Common Fisheries Policy (article 15 of the EU regulation No 1380/2013) since 2019 [16], restricts the practise of discarding and requires fishers to land their entire catch of species under TAC management, including skates. Because landing skates results in 100% mortality, species with scientific evidence for a high survival probability when discarded are eligible for exemptions to the landing obligation. Currently all fisheries in Western waters [18] and the North Sea [19] are exempted from the obligation to land their entire catches of skates that fall under the common TAC [15]. The exemptions [18, 19] do not specify the species and expire in 2027.

Fishers have a clear interest in these exemptions because they mitigate the risk of early depletion of quota and subsequent closures of fisheries when catches of so called 'choke species' cannot be avoided. Because of the relatively small TAC, skates are particularly prone to become choke species. Marine nature conservationists also support exemptions to the LO to reduce unnecessary fishery mortality among skates while favouring population recovery [20], especially because their survival probability is expected to be relatively high [21].

Post-capture survival and condition of skates varies with variables such as fishing gear, species, size and temperature [e.g. 5, 22–24]. Carefully designed discards survival studies are needed to obtain results that are representative for fishing gears, species and conditions of main interest. In case of Dutch demersal fisheries, beam trawl fisheries targeting sole and flyshoot fisheries, also known as Scottish seine fisheries, are the main contributors to both skate discarding and landings, with thornback and spotted skate as main species [25]. Discards survival probability measurements in Dutch demersal fisheries are thus most relevant for these gears and species. Previous studies on skate discards survival in North Sea beam trawl fisheries reported short-term survival (60–120 h post catch monitoring, e.g. [26–28]) and this may be too short to determine the fate of the sampled fish [29]. Van Bogaert et al. [22] monitored survival of a limited number of thornback skates (n = 21) up to 21 days post catch and reported a survival probability of 56.9%. Survival probabilities of skates discarded by flyshoot fisheries have not been measured.

The objectives of this study were to measure survival probabilities of skates discarded by tickler chain and pulse beam trawl fisheries and flyshoot fisheries and to explore which

variables predict survival probability. We conducted 18 trips during which in total 397 thornback skates and 161 spotted skates were sampled from commercial catches. Survival probabilities were determined by captive observations of the sampled skates for 15 to 25 days post catch.

## Materials and methods

### Ethics statements

The treatment of the fish was in accordance with the Dutch animal experimentation act, as approved by ethical committees (Experiments, 2017.D-0012.002, 2018.D-0002.004 and 2021.D-0007.001). Humane end points nor alleviations of suffering were applied as these would interfere with survival time measurements. The methodology was in accordance with the guidelines for discards survival studies developed by the Workshop on Methods for Estimating Discard Survival (WKMEDS) of the International Council for the Exploration of the Sea (ICES) [29]. Release of surviving skates to the Eastern Scheldt after the completion of experiments was in accordance with IUCN (International Union for Conservation of Nature) Guidelines for reintroductions and other conservation translocations [30].

### Experiments

**Trips.**   Test-fish were collected during eighteen regular commercial fishing trips on three commercial pulse trawlers (Pulse trawl, 9 trips, Tables 1 and S1), one tickler chain beam trawler (Beam trawl, 5 trips, Tables 1 and S2) and a commercial flyshooter (Flyshoot, 4 trips, Tables 1 and S3) between 2017 and 2023. All trips lasted between four and five days, comparable to commercial trip lengths. All fishery operations were conducted in the southern North Sea and the English Channel according to commercial practices of the fishing vessels. Trips were spread out over the year (Table 2) to account for the potential effect of variable environmental and fishing conditions on discards survival [23, 32]. Operational and environmental conditions during sea trips were recorded for each haul on trawl lists by the skippers (Table 2).

**Test-fish collection.**   In total, 557 test-fish were sampled with thornback skates ranging from 17 to 92 cm and spotted skates ranging from 24 to 63 cm total length (TL). Although landings are currently restricted to skates with total lengths above 45 to 55 cm, larger skates were not excluded from the samples because quota limitations may force fishers to discard larger specimens. All skates were randomly sampled during the semi-automatic catch-sorting process which is common in these fisheries. In this process, catches were discharged from the cod-end(s) into one (flyshoot) or two (pulse and beam trawl) hoppers. From the hoppers, the catches were flushed into a central pit from which the catch was transported by a conveyer belt onto the sorting belt from which marketable fish were manually collected by crew members. At the end of the sorting belt, the remaining unwanted catch, including fish with no commercial value and undersized fish, dropped into a gutter with a water supply that discharged the catch back into the sea.

**Table 1. Fishing vessels used.**

| Vessel | 1 | 2 | 3 | 4 | 5 |
|---|---|---|---|---|---|
| Gear | Pulse trawl | Pulse trawl | Pulse trawl | Beam trawl | Flyshoot |
| Engine power (Kw) | 1471 | 1430 | 1470 | 1888 | 680 |
| Tonnage (GT) | 426 | 366 | 494 | 494 | 340 |
| Length (m) | 41 | 39 | 42 | 42 | 31 |
| Fishing speed (kn) | 4.8 | 4.8 | 4.9 | 6.1 | n.a. |

**Table 2. Conditions during the sea trips.**

| Trip | Gear | Vessel | Year | Month | Week | Mean water temp. (˚C) | Wind speed range (Bft) | Mean wave height (m) | Mean catch processing (min) | Mean haul duration (min) | Mean fishing depth (m) |
|---|---|---|---|---|---|---|---|---|---|---|---|
| 1 | Pulse trawl | 1 | 2017 | May | 18 | 9–12 | 2–5 | 0.5–2.0 | 30 | 110–135 | 18–28 |
| 2 | Pulse trawl | 2 | 2017 | May | 21 | 12–13 | 0–4 | 0.2–0.5 | 24 | 120 | 30–50 |
| 3 | Pulse trawl | 3 | 2017 | June | 24 | 14–15 | 1–5 | 0.1–1.5 | 20 | 110–125 | 22–24 |
| 4 | Pulse trawl | 3 | 2017 | July | 28 | 16–17 | 1–6 | 0.1–1.0 | 23 | 110–120 | 25–40 |
| 5 | Pulse trawl | 1 | 2017 | Sept | 36 | 18 | 4–5 | 0.5–1.5 | 26 | 120–145 | 26–37 |
| 6 | Pulse trawl | 3 | 2017 | Oct | 44 | 13–15 | 3–5 | 0.5–2.0 | 20 | 110–130 | 27–34 |
| 7 | Pulse trawl | 2 | 2017 | Dec | 49 | 11–12 | 3–5 | 1.0–2.0 | 34 | 120 | 35–50 |
| 8 | Pulse trawl | 1 | 2018 | Jan | 4 | 6–7 | 5–6 | 0.5–2.6 | 33 | 120 | 28–39 |
| 9 | Pulse trawl | 2 | 2018 | Feb | 8 | 7–8 | 2–5 | 0.5–1.5 | 25 | 110–120 | 22–52 |
| 10 | Beam trawl | 4 | 2021 | Oct | 41 | 15.9 | 3–4 | 1.14 | 17 | 126 | 27 |
| 11 | Flyshoot | 5 | 2021 | Nov | 47 | 13.5 | 2–5 | 1.1 | 44 | 88 | 32 |
| 12 | Beam trawl | 4 | 2022 | Jan | 3 | 8.7 | 1–4 | 1.7 | 28 | 126 | 28 |
| 13 | Flyshoot | 5 | 2022 | Mar | 13 | 10.0 | 2–4 | 1.3 | 24 | 69 | 45 |
| 14 | Beam trawl | 4 | 2022 | May | 19 | 11.3 | 1–4 | 0.7 | 24 | 124 | 27 |
| 15 | Flyshoot | 5 | 2022 | Jun | 24 | 16.0 | 1–4 | 0.5 | 26 | 67 | 38 |
| 16 | Beam trawl | 4 | 2022 | Jun | 26 | 15.6 | 1–3 | | 23 | 124 | 27 |
| 17 | Flyshoot | 5 | 2022 | Aug | 33 | 20.2 | 0–4 | 1.0 | 24 | 67 | 37 |
| 18 | Beam trawl | 4 | 2022 | Sep | 38 | 18.1 | 1–6 | 0.9 | 24 | 121 | 26 |

Trips with pulse trawlers were dedicated to discards survival measurements for a variety of species including thornback skate. We aimed for balanced sampling of catch processing of three skates per haul but this was occasionally hampered by availability of skates in the catches. The total number of sampled skates per pulse trawl trip ranged from nine to fourteen specimens from three to seven hauls per trip.

For the beam trawl and flyshoot trips, that were fully dedicated to discards survival measurements of skates, the total of 40 skates < 65 cm TL and 6 > 65 cm TL that could be sampled per trip was determined by the housing capacity in the on-board monitoring units. Note that the cut-off at > 65 cm TL was dictated by the tank size of the type 1 on-board monitoring units (see below) and is unrelated to the MLS of skates. We aimed to sample 20 thornback and 20 spotted skates smaller than 65 cm TL each trip by sampling five skates from four hauls for both species. Each sampled haul we aimed to sample the first two and last three skates per species that appeared on the sorting belt to account for the potential effect of time spent in catch processing on survival probability. Low numbers of skates in some of the catches incidentally forced us to deviate from this sampling scheme. Occasionally, skates were sampled from the mid part of catch processing to obtain the desired sample size per haul. For each sampled haul the start and end time of catch sorting were recorded to obtain the approximate sampling time of individual skates. Each beam trawl and flyshoot trip large thornback skates (> 65 cm TL) were opportunistically sampled according to their availability in catches. Spotted skates > 65 cm TL were not available in any of the sampled hauls. We sampled more skates per trip than aimed for in case skates were dead when sampled or died at sea and when two small skates (< 25 cm) could be housed in one tank. All sampling was random and not influenced by the condition of skates in the catches. Samples were subsets of the total amount of skates appearing on the sorting belts and we only recorded data for these subsets.

**Control fish.** During all trips, control fish of the same species as the test-fish (Trips 1 to 9, n = 2; Trips 10 to 18 n = 4 per species) were used to separate potential effects of the

experimental procedures on mortality from fisheries-induced mortality. At the start of a trip, control fish were stored on deck in aerated 600L tanks. At sea the tank water was regularly renewed with surface seawater. Control fish were exposed to the exact same experimental procedures throughout the experiments as test-fish from the moment of test-fish collection from the sorting belt. Control fish were obtained by collecting the least damaged skates from the catches of a research vessel and by using test-fish from previous trips as control fish. Prior to their use in experimental trips, control fish were stored in tanks placed in a climate-controlled room for at least three weeks. During this period, fisheries-induced mortality levelled out while surviving fish could recover from injuries and regain good condition. During storage, fish were fed daily with dead, uncooked brown shrimps (*Crangon crangon*) and pieces of herring (*Clupea harengus*) and Atlantic mackerel (*Scomber scombrus*) to visually observed satiation. Only fish in visually observed good condition, well-fed and without visible injuries, were used as control fish.

**Condition assessment and monitoring of survival.** Skates sampled from catches were temporarily stored in 105L holding containers filled with seawater. Seawater in holding containers was regularly renewed to maintain dissolved oxygen levels during storage. Once sampling of a haul had been completed, skates were sequentially taken from the holding containers for condition assessment, to measure total length (TL: in cm below), to determine their sex and for tagging. Condition of each individual skate was determined and scored A to D by scoring reflex impairment and damages, using a methodology developed for flat fish [23] that was adapted to skates after [22] (Table 3). Skates were tagged with Trovan Unique glass transponders (type ID100) to allow for identification of individuals. Transponders were injected subcutaneously at the base of the tail using an IID100E injector. Upon completion of their initial assessment, skates were individually placed in 24 L tanks (TL < 65 cm) or 84 L tanks (TL > 65 cm). Occasionally two small specimens (TL < 25 cm) were placed in a single 24 L tank that was split into two equally sized compartments by a perforated plexiglass sheet. Sampled skates that showed no spiracle movement for more than 15 seconds during condition assessment were recorded as dead at time zero and not placed in tanks. Survival monitoring started at sea and continued in the laboratory. All tanks were inspected for mortalities every 12 hours on board and every 24 hours in the laboratory. Dead fish were detected by visual confirmation of the absence of spiracle movement, immediately removed from their tanks, and identified by their transponders. The date and time at which skates were found dead was recorded. Lethargic fish were not removed as for their potential recovery and to obtain actual survival time. Dissolved oxygen concentration and water temperature were measured (Hach Lange Multimeter). Water flows to individual tanks were increased if oxygen saturation was below 80%. All experiments were terminated after 14 (trip 1 to 9) or 21 (trip 10 to 18) days of survival monitoring in the laboratory. Depending on the day of sampling at sea, this resulted in post catch monitoring periods for surviving individuals ranging between 15 and 18 days for trip 1 to 9 and 22 and 25 days for trip 10 to 18. Upon termination of the experiment, all surviving skates were netted from the tanks and identified by their transponders to record species, total length (TL: in cm below) and sex. Surviving skates were either released to the Eastern Scheldt or kept in captivity for use as control fish in subsequent survival experiments.

## Experimental facilities

Two types of custom-built monitoring units were installed on board the fishing vessels to house test-fish and control fish. Type 1 units consisted of a stainless-steel framework holding 16 polyethylene 24L tanks (60 cm L x 40 cm W x 12 cm H) suitable for skates with a total length < 65 cm. The type 2 unit held six polyethylene 84 L tanks (80 cm L x 60 cm W x 17.5 cm H) to house

**Table 3. Description of criteria to score condition and determine vitality class of skates.** Methodology for flat fish by Van der Reijden et al. [31] modified for skates after Van Bogaert et al. [22].

| Vitality class | Description |
|---|---|
| A | Fish lively, no visible or very minor external damages. |
| B | Fish less lively, minor scratches or discolorations on up to 20% of the dorsal skin surface area, some haemorrhaging on the ventral side, no or minor bruises on the ventral side. |
| C | Fish lethargic, intermediate scratches or discolorations on up to 50% of the dorsal skin surface area, several haemorrhaging and/or bruising on the ventral side. |
| D | Fish lethargic or dead, major scratches or discolorations on the dorsal skin surface area, significant haemorrhaging and/or bruising on the ventral side. |
| **External damage scores** | Description (scoring: 1 = present; 0 = absent) |
| Wings | Wings are damaged or split. |
| Dorsal side | Damage to skin surface, scratches or discolorations at dorsal pigmented body surface. |
| Hypodermic haemorrhages | Superficial hypodermic haemorrhage on the ventral white body surface |
| Hypodermic bruises | Hypodermic purple-reddish bruise on the ventral white body surface |
| Intestines | Intestines are protruding or are visible through damaged body tissue of the fish. |
| Wound | A wound such that flesh is visible anywhere on the body including torn off thorns. |
| **Reflex impairment scores** | Description (1 = impaired; no (clear) response within 5 s of observation; 0 = unimpaired; obvious response within 5 s). |
| Wings | Skate is held out of the water, dorsal side up with one hand supporting the body at the head of the skate and the other hand supporting the body at the base of the tail. The skate actively flaps its pectoral fins (wings). |
| Eye retraction | While out of the water the skate is gently tapped on the head just behind the eyes with a blunt probe. The skate actively retracts its eyes. |
| Tail grab | While in the water resting on the tank bottom the skate is gently held by the tail. When the tail is gently pulled backwards, the skate struggles free and swims away (or attempts to do this). |
| Spiracles | While in the water resting on the tank bottom the movement of the spiracles is observed. |

six skates with a total length > 65 cm. During trips 1 to 9 (pulse trawl trips) four type 1 units were installed on board, resulting in a total capacity of 64 tanks, of which nine to fourteen were available for skates, while the other tanks were used to house flat fish species (not reported on here). Trips 10 to 18 (beam trawl and flyshoot trips) each employed three type 1 units and one type 2 units, resulting in a total capacity to house 48 skates < 65 cm TL and 6 skates > 65 cm TL. In all units, each tank was equipped with an individual water supply. A pump with a water intake approximately 2 meters below sea surface continuously supplied seawater to the tanks. Water flow rates were approximately two tank volumes per hour to maintain proper water quality. Tanks were covered with transparent lids to limit water loss by sloshing while allowing for visual inspection of fish. Upon return in port, the units were transported to the laboratory in a temperature-controlled truck. Transport time ranged from one to four hours depending on the home port of the vessel. During transport, each unit was placed inside a tank that was partly filled with seawater and equipped with a submerged pump to supply water to each fish tank in the unit. Fish tanks discharged their effluents in the tank in which the unit was placed, allowing for recirculation and aeration of the water. Upon arrival at the laboratory, fish were removed from their tanks and housed in a tank system placed in a temperature-controlled room. Skates > 65 cm TL were housed in tanks with a bottom surface area of 2.5 m$^2$ at a maximum density of 4 fish per tank (1.6 m$^2$/fish). Skates < 65 cm TL were housed in tanks with a bottom surface area of 2 to 2.5 m$^2$ at a maximum density of 8 fish per tank (0.5 to 0.8 m$^2$/fish). Control fish were mixed with test fish. Different species were housed in separate tanks. All tanks were

connected to a single water recirculation system with a biological filter. Water in the system was continuously renewed with filtered seawater from the Eastern Scheldt at a rate of 2 to 3 m$^3$/d. In the laboratory, all tanks were supplied with coarse sand as bottom substrate and fish were fed daily to visually observed satiation with uncooked brown shrimps (*Crangon crangon*) and pieces of herring (*Clupea harengus*) and Atlantic mackerel (*Scomber scombrus*).

## Data analysis

**Survival curves.**   To visualise and analyse the survival of sampled fish, Kaplan-Meier survival curves [33] were fitted and 95% Confidence Intervals were computed using *survfit* function from the *survival* package [34] in R [35]. Survival curves and their 95% Confidence Intervals were created for control fish vs. test-fish by gears and species and for test-fish by vitality class. Fish that died due to unnatural causes other than fisheries, mainly by jumping from the containment tank, were censored: this means that rather than being registered as dead, their experiment was said to have been terminated at the time of death.

**Statistical tests survival.**   To test if differences in survival were significant between species, gears and vitality classes, the functions *survdiff* and *pairwise_survdiff* from the *survival* package were used, which take the times until death (or censoring) and a categorical group and computes a test statistic based on the log-rank test [36]. The log-rank test statistic was calculated by comparing the observed and expected number of events in each group at various time points throughout the study period. It measured the discrepancy between observed and expected events under the assumption that survival experiences for all groups were identical. This was done to detect differences in discards survival probabilities of thornback and spotted skate between gears and vitality classes. Effects were considered significant when p < 0.05.

**Predictors of survival.**   Generalized Linear Mixed-Effect Models (GLMMs) were computed to identify the combination of variables that best explained the variation in discard survival, using the *glmer* function from the *lme4* package in R [37]. In these models, the haul from which fish were sampled (incorporating the variables vessel and trip) was included as random effect. Models of all combinations of the explaining variables gear, species, length, catch processing time, sea surface temperature, depth and wave height, including all possible interactions were computed using the *dredge* function from the *MuMIn* package in R [38]. Haul duration was not considered because variation was low and because the haul process is different between (pulse) beam trawling and flyshooting, meaning haul duration cannot be considered analogous between the different fishing methods. Wind speed was not considered because of its strong correlation to wave height. Sediment type was not considered because most observations occurred on the same sediment type (sandy). Numerical variables were normalized to increase the performance of the models. The model with the lowest Akaike Information Criterion (AIC) value was considered the optimal model.

**Vitality class as predictor of survival.**   To investigate the effectiveness of vitality class a predictor of discard survival, GLMMs were computed employing the selected combination of variables that best explained variation in discards survival, as well as vitality class. Vitality class was considered a generic predictor of survival in case it was the only variable selected in the GLMM with the lowest AIC.

The presence of observer bias [39] in the vitality classification was studied using a Cumulative Link Mixed Model (CLMM) with vitality class as dependent variable and observer as explaining variable. Using the *drop1* function from the *stats* package in R, we performed a hypothesis test based on the chi-squared distribution to assess the significance of dropping the observer variable from the model. Observer bias was present in case the model with observer as explaining variable was significantly better at predicting vitality class than the null model (p < 0.05).

## Results

### Discards survival probabilities

For all gear types and species, survival probability estimates were based on trips spread out over the year and conducted under variable environmental conditions (Table 2). Survival probability estimates for thornback skate differed among gears (p<0.001, Table 4). The discards survival probability observed for thornback skate in flyshoot fisheries was higher than for beam trawl and pulse trawl fisheries (p < 0.001). No difference in survival probability of thornback skate was detected between beam and pulse trawl fisheries (p = 0.48). Survival probability estimates for spotted skate differed between flyshoot and beam trawl fisheries (p = 0.02, Table 4). For pulse trawl fisheries no spotted skates were sampled.

For both flyshoot and beam trawl fisheries most of the mortality among spotted skates occurred within the first two to five days of the experiment and levelled out thereafter (Fig 1B and 1E). Mortality among thornback skates occurred up to approximately day 20 for beam trawl fisheries (Fig 1A), up to day 15 for pulse trawl fisheries (Fig 1C), and up day 10 post catch for flyshoot fisheries (Fig 1D) before levelling out. Direct mortality, i.e. dead at-vessel when sampled, just before the moment skates would have been discarded, was lowest for pulse trawl fisheries (Table 4). Only one out of a total of 86 control fish died in the experiments, resulting in control fish survival ranging from 94% for pulse trawling to 100% for the other two gears.

### Predictors of survival

Survival was best explained by the GLMM including the variables gear, water temperature, catch processing time, wave height and the interaction between wave height and water temperature. Variables fishing depth, length and species were not selected. Table 5 presents the estimates and their significance of the fixed effects. All selected variables have a significant effect on discards survival probability except for the gear effect of pulse trawling, i.e. survival probability does not differ between pulse and beam trawling. Fig 2 gives the probability distributions with 95% confidence intervals for the effects of the selected variables on survival probability at each value of the fixed effect with the other variables set to the mean value (or mode for non-numeric variables). Survival probability differed among gears (Fig 2A) with higher survival for skates discarded by flyshoot fisheries (p < 0.001) but no difference between pulse and beam trawl fisheries (p = 0.58). Survival probability decreases with catch processing time (Fig 2C, p = 0.024) and to a lesser extent with wave height (Fig 2D, p = 0.02) and water temperature (Fig 2B, p = 0.027). The effect of water temperature depends on wave height (Fig 2E, p = 0.022).

**Table 4. Discards survival probability estimates (%) and their 95% confidence intervals, direct mortality and survival of control fish for spotted and thornback skate discarded by beam trawl, pulse trawl, and flyshoot fisheries.** Survival probability estimates within species and per gear with different letters in superscript are significantly different. Note that no comparisons between species were made.

| Species | Gear | Survival probability (95% CI) | n | Direct mortality | Survival of control fish | n |
|---|---|---|---|---|---|---|
| Thornback skate | Beam trawl | 50% (43–57)[a] | 183 | 11.8% | 100% | 21 |
| | Pulse trawl | 54% (40–65)[a] | 94 | 2.1% | 94% | 18 |
| | Flyshoot | 80% (73–87)[b] | 120 | 4.0% | 100% | 16 |
| | p-value | p <0.001 | | | | |
| Spotted skate | Beam trawl | 44% (37–54)[a] | 136 | 11.3% | 100% | 19 |
| | Flyshoot | 75% (60–94)[b] | 24 | 2.5% | 100% | 12 |
| | p-value | p = 0.02 | | | | |

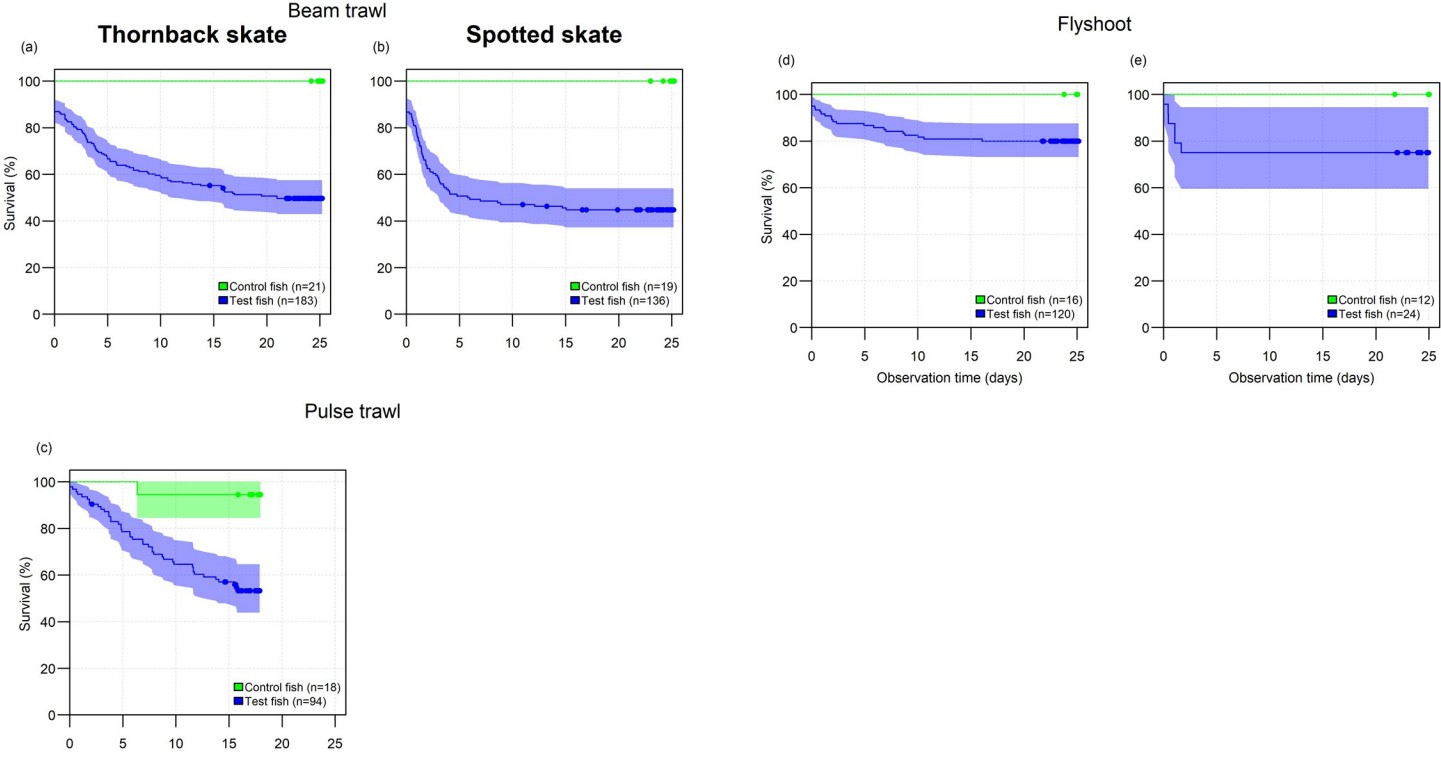

**Fig 1.** Kaplan-Meier survival curves for thornback (a,c,d) and spotted skate (b,e) discarded by beam trawl (a, b), pulse trawl (c) and flyshoot fisheries (d, e). Curves are plotted for control fish and test-fish employing all fish sampled during five beam trawl trips, nine pulse trawl trips and and four flyshoot trips. Drawn lines indicate mean survival (percentage over time), with shaded areas indicating 95% confidence intervals. Dots indicate the end of the monitoring time for individual fish that were alive at the end of the experiments.

## Vitality class as predictor of survival

The condition in which skates were landed on deck was expressed by vitality classes A to D. Vitality class had a significant effect on discards survival probability (Table 6) and for all species-gear combinations survival probability declined with deteriorating condition. In all cases survival probability was highest for vitality class A and lowest for vitality class D, but not all observed survival probabilities differed among vitality classes (Table 6). GLMMs employing the previously selected variables that best explain survival probability (gear, catch processing time, water temperature, wave height) plus vitality class were computed to investigate to what extend vitality class is a generic predictor of survival probability. Survival was best explained

**Table 5. Estimates for the fixed effects and their significance in the Generalized Linear Mixed-Effect Model explaining the survival probability of thornback and spotted skate discarded by beam trawl, pulse trawl and flyshoot fisheries.** Gear effects are relative to beam trawl fisheries. Significant fixed effects are marked with *.

| Fixed effects: | Estimate | Std. Error | z value | Pr(>|z|) |
|---|---|---|---|---|
| Intercept | 2.1118 | 0.8303 | 2.543 | 0.011 * |
| Gear Flyshoot | 2.0018 | 0.3875 | 5.166 | <0.001* |
| Gear Pulse trawl | 0.1991 | 0.3627 | 0.549 | 0.580 |
| Wave height | -4.5603 | 1.9667 | -2.319 | 0.020 * |
| Water temperature | -2.4954 | 1.1306 | -2.207 | 0.027 * |
| Catch processing time | -2.7659 | -1.2236 | -2.260 | 0.024 * |
| Water temperature*Wave height | 6.1802 | 2.6887 | 2.299 | 0.022 * |

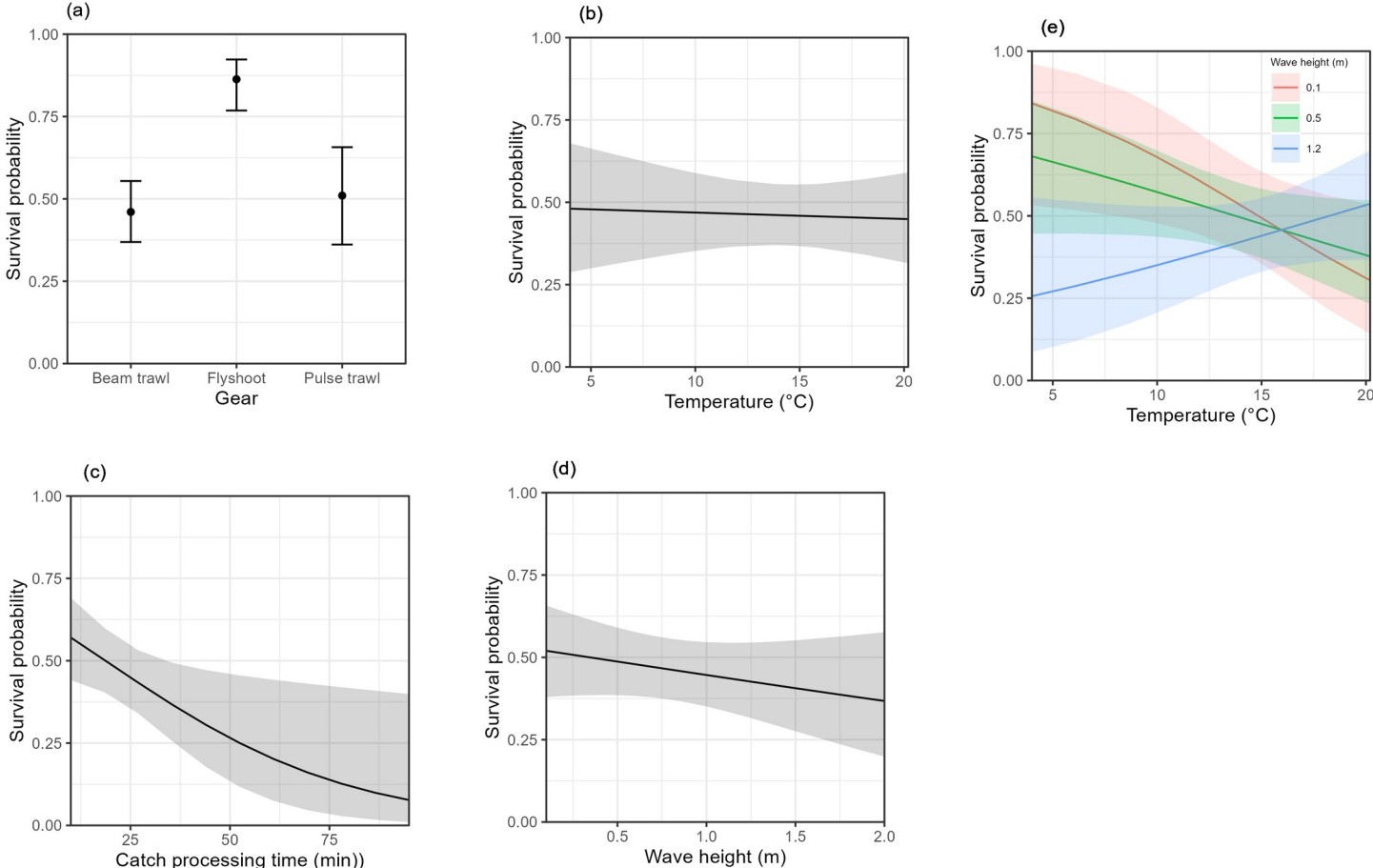

**Fig 2. Predicted probabilities of discards survival for the variables in the GLMM with the lowest AIC value.** Variables selected in the model were a) gear, b) water temperature, c) catch processing time, d) wave height and e) the interaction between water temperature and wave height. The graphs represent the probability of survival at each value of the fixed effect with the other variables set to the mean value (or mode for non-numeric variables).

(lowest AIC) by a model that included vitality class, gear, water temperature and the interactions between gear and water temperature and between temperature and vitality. Apparently, vitality class incorporates the effects of catch processing time and wave height on survival as these variables were no longer selected. Kaplan-Meier survival curves by vitality class reveal distinct differences in survival between vitality classes for beam and pulse trawl fisheries (Fig 3A) while for flyshoot fisheries vitality classes A, B and C show more similar survival curves (Fig 3B). This, combined with the selection of the gear and water temperature in the model, reveals that vitality class is not a generic predictor of skate discards survival. No observer bias

**Table 6. Survival probability estimates (%) and their 95% confidence intervals per vitality class of spotted and thornback skates discarded by beam trawl, flyshoot and pulse trawl fisheries.** The letters in superscript refer to the vitality classes from which the survival probability is significantly different (within gear and species).

| Species | Gear | A | B | C | D | p-value |
|---|---|---|---|---|---|---|
| Spotted skate | Beam trawl | 100% (100–100)[C,D] | 74% (64–86)[C,D] | 19% (8–41)[A,B,D] | 5% (1–19)[A,B,C] | p < 0.001 |
| | Flyshoot | 100% (100–100)[D] | 73% (54–100)[D] | 80% (52–100)[D] | 0%[A,B,C] | p < 0.001 |
| Thornback skate | Beam trawl | 97% (90–100)[D] | 70%(60–81)[C,D] | 50% (37–66)[B,D] | 12% (6–25)[A,B,C] | p < 0.001 |
| | Flyshoot | 97% (90–100)[D] | 90%(83–98)[D] | 79% (65–95)[D] | 39% (22–69)[A,B,C] | p < 0.001 |
| | Pulse trawl | 90% (80–100)[C,D] | 67% (53–84)[C,D] | 44% (30–65)[A,B,D] | 8% (1–54)[A,B,C] | p < 0.001 |

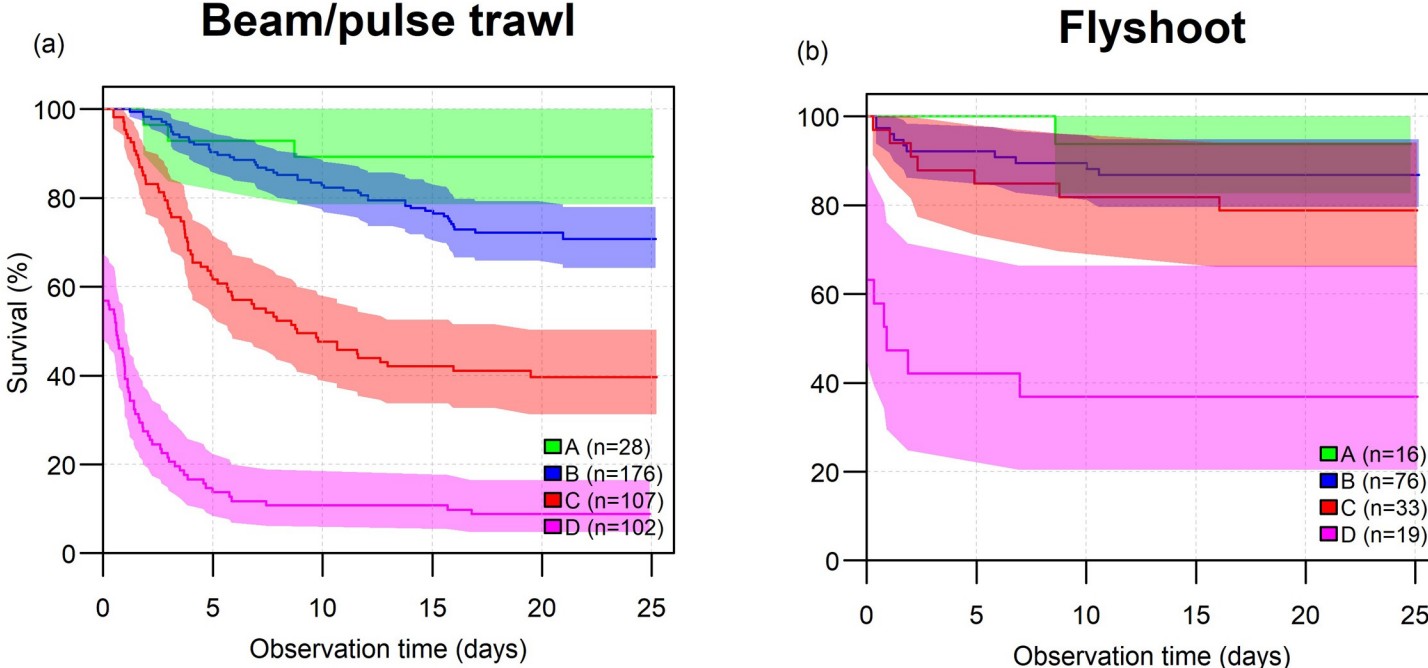

**Fig 3.** Kaplan-Meier survival curves for test-fish of vitality classes A to D discarded by beam and pulse trawl (a) and flyshoot fisheries (b) combining the observations for thornback and spotted skate. Note that vitality class D includes fish that were dead at the start of the experiment. Drawn lines indicate mean survival percentage over time, with shaded areas indicating 95% confidence limits.

in vitality classification was detected; the CLMM with observer as explaining variable was not significantly better than the null model (p = 0.44).

## Discussion

The main objective of this study was to provide estimates for the survival probabilities of skates discarded by beam trawl, pulse trawl, and flyshoot fisheries. To this end we conducted 18 trips spread throughout the year, during which 397 thornback skates and 161 spotted skates were sampled from commercial catches. Sampled skates were housed in tanks to monitor their survival over time. The artificial conditions and fish handling associated to captive observation experiments may cause additional mortality through additional stress [29] and thus underestimate actual discards survival. In our experiment however, we consider all mortality among sampled skates to be fisheries induced because all but one control fish, used to distinguish mortality caused by the experimental procedures from fisheries-induced mortality, survived. Captive observation may also result in overestimation of actual discards survival because it excludes the contribution of predation to discards mortality [40]. Skates recovering on the seafloor may be prone to mortal injury by bottom scavengers [41]. Because of the unknown level of predation related mortality, the discards survival probability estimates reported here should be considered as maximum values.

To the best of our knowledge, this study presents for the first time discard survival probabilities for skates discarded by flyshoot fisheries. Consequently, there are no previous studies to which we can compare our results. There are only a few other studies on skate discards survival in North Sea beam trawl fisheries. Apart from the study by Van Bogaert et al. [22], these studies reported short-term survival. Short-term survival (80h) of 72% was observed for unspecified skate species (given the sampling area probably at least partly thornback and spotted

skates) sampled from catches of eurocutters with chain mat beam trawls [28]. Kaiser and Spencer [26] reported 59% survival after 5 days of monitoring for cuckoo skate (*Leucoraja naevus*) caught by a 4 m beam trawl towed for 0.5 h. Enever et al. [27] reported 59% short term (60h) survival for thornback skates caught in commercial beam trawl tows in the Bristol Channel. Because captive observation requires a sufficiently long monitoring period to determine the fate of sampled fish [29], and because our survival curves reveal that, especially for thornback skate, mortality levels out only after 15 days, it is likely that short-term studies reported overestimated survival probabilities. We monitored survival for 15 to 18 days post catch in the first skate discards survival experiments for pulse trawl fisheries and extended the monitoring period in the laboratory to 21 days in later experiments for flyshoot and beam trawl fisheries. This resulted in survival monitoring for 22 to 25 days post catch for individual skates depending on the day of sampling at sea. Despite the extended monitoring period, survival probability of thornback skate potentially decreased further for beam trawl fisheries after the experiments were stopped. We therefore consider our estimate of discard survival probability for thornback skate caught by beam trawl fisheries a maximum value. We recommend a minimum monitoring period of 25 days post catch for thornback skate in future research.

Mortality among fish caught and discarded by commercial fisheries is caused by failure to recover from fisheries induced stressors such as hypoxia, injury, exhaustion, barotrauma and predation to which these fish are exposed during capture, handling and release [42, 43]. The effects of gear, catch processing time, water temperature and wave height on skate discards survival that we observed, can therefore be attributed to the effects of these variables on the severity of fisheries induced stressors.

The higher survival that we observed in flyshoot fisheries, can be explained (i) by the shorter time that fish are retained in the cod-end in these fisheries (maximum 20 minutes) compared to pulse and beam trawling (up to 120 minutes), and (ii) by the smaller amounts of benthic organisms and debris in the catches of flyshoot fisheries. Both result in lower mechanical impact, less injuries and exhaustion, compared to beam and pulse trawling. Surprisingly we did not detect a difference in survival probability between pulse and beam trawl fisheries. We predicted a lower survival in beam trawling because of the higher towing speed of tickler chain beam trawls (6–7 kn) and generally larger catch volumes with more benthic organisms and debris compared to pulse beam trawls (~ 5 kn). Higher towing speeds may lead to lower discards survival through faster exhaustion, more injuries due to more severe collisions with the net, other fish and debris, more dense crowding and increased compression of fish in the cod-end.

It should be noted that within gear variation in discard survival may exist between individual vessels due to differences in fishing and catch processing procedures. Disentangling vessel effects from effects of inevitably varying sea conditions among trips requires a very large number of sampled trips. Differences in fishing and catch processing procedures are probably much larger between gears than among vessels that use the same gear. Therefore, we attribute the difference in discards survival probabilities observed between beam trawl and pulse trawl fisheries on the one hand and flyshoot fisheries on the other hand to a gear effect rather than a vessel effect. Also, we consider the survival probability estimates representative for the gears.

As discards survival probability declines with increased duration of hypoxic conditions [42], it is not surprising that survival probability declined with increasing catch processing time, i.e. increasing air exposure time. During air exposure, oxygen deficiency in tissues generally occurs due to collapsed gill lamellae which drastically reduces the fish' gas exchange capacity [42]. It should be noted however that effects of variables catch processing time, catch mass and haul duration are closely related: longer hauls lead to larger catches which lead to longer catch processing time. Each variable may affect discards survival by itself or by its effect on another variable. For example, decreasing survival with increasing haul duration, reported by

e.g. [44], has been attributed to longer exposure to the mechanical impacts, i.e. compression [23] imposed on fish in the trawl as well as the increased catch mass resulting from longer tows [27, 45]. Increased catch mass itself leads to denser crowding which exacerbates bruising, crushing and constriction injuries as fish are pushed against the cod-end mesh, other biota and debris [46]. This collinearity makes it is difficult to attribute effects on survival to individual variables. Unfortunately, we lack both catch mass data (not measured) and useful haul duration data (no contrast due to sampling from regular, unmanipulated commercial fisheries) to explore their effects on survival probability of discarded skates. We therefore cannot exclude that effects of catch mass contributed to the observed effect of catch processing time on survival probability.

Provided that air exposure during catch processing contributed to skate discard mortality, survival may be increased by discharging catches in water filled hoppers instead of dry hoppers. Well designed and managed water filled hoppers in which the fish do not deplete dissolved oxygen levels and do not sustain injuries due to collisions caused by sloshing during rough seas, may reduce the impact of hypoxia because it shortens air exposure time during catch processing. Plaice discards survival did not benefit from water filled hoppers because most of the plaice caught with pulse beam trawls were already lethally damaged in the trawl prior to being discharged in the hoppers. This also explains why survival of these plaice was unaffected by catch processing time [32]. In case of skates however, our observation that catch processing time does matter suggests that in contrast to plaice, skate discards survival may benefit from water filled hoppers.

Lower survival probability with increasing surface water temperature, as we found for discarded skates, has been previously reported for chondrichthyans [47] as well as teleost fish, e.g. [32, 48]. This temperature effect can be attributed to an increase in physiological stress at higher temperatures and exposure to temperature change [42]. Whether the current temperature effect was caused by the higher temperature itself or exposure to temperature change, which is largest during summer months [49], is not clear. The temperature effect seems rather small at first. However, the potential effect of temperature on survival is revealed by the interaction between water temperature and wave height. When waves are low, water temperature has a much stronger negative effect on survival probability than when temperature is considered as a main effect with all other variables in the model are set at their average values. The high survival at low water temperature when seas are calm is strongly reduced as soon as wave height increases. This effect of sea state is probably caused by an increase in mechanical impact of the gear on the fish [42] overruling the temperature effect. There is no logical explanation for a positive effect of wave height on survival at higher water temperature and we consider this model result an artifact of the limitations of the GLMM. More complex models seem to be needed to describe this interaction at high water temperatures.

In contrast to Van Bogaert et al. [22] we found no significant difference in survival probability between thornback and spotted skate. Numerically however and in agreement with Van Bogaert et al. [22] we observed the highest survival for thornback skate. Thornback skate's more rugged and spinulose external surface has been suggested to offer better physical protection to mechanical impacts of capture processes and thus higher survival chances when discarded compared to more smoother skate species such as blonde (*Raja brachyura*) and spotted skate [27]. This morphological advantage of thornback skate seems to turn into a disadvantageous higher tangling risk when caught in fixed nets, resulting in lower immediate survival and higher reflex impairment and injury compared to other skate species [23]. It thus seems that species differences in discards survival can be fishing gear dependent.

We found no evidence for a length effect on survival probability. This is surprising because various studies reported size-dependency of survival probability. For example, both Van

Bogaert et al. [22] and Depestele et al. [28] reported that smaller specimens have lower survival chances while Clarke et al. [23] reported lower immediate survival and vitality for larger skates. As Clarke et al. [23] pointed out, more insight in size-dependency across gear types and species could inform species-specific, and in our view also fishing gear-specific, MLS. We currently have no explanation why we did not find a length effect but it seems unlikely that it is caused by a lack of length variation in our data that cover a wide length range.

All test-fish were assigned a vitality class A, B, C or D based on individual condition assessments directly after sampling. Survival probabilities among test-fish grouped by vitality class differed across species and gears. Without exception vitality classes A and D always yield the highest and lowest discards survival probabilities. Vitality class thus proved to be a fairly good predictor for skate survival, similar to flatfish [39]. Also, it is clear that the condition in which the fish arrive on the sorting belt has a strong effect on their survival chances when discarded. Survival chances can thus be increased by measures that reduce deterioration of fish condition prior to their landing on deck. The effect of vitality class on survival was most pronounced for skates caught by beam and pulse trawl fisheries and less in flyshoot fisheries. Also, survival by vitality class was higher in flyshoot fisheries, which suggests that in beam and pulse trawl fisheries skates died due to injuries that were not picked up by the condition assessment we used. In addition, other variables next to vitality class were selected when computing GLMMs to predict survival from vitality class. We therefore doubt that vitality classes as we assigned them, are generic predictors of skate discards survival and that accurate prediction of survival based on our condition assessment is possible. We consequently did not attempt to construct a mathematical model that predicts of discards survival using vitality class and the underlying reflex impairment and damage scores. For the moment, assigning vitality classes seems a useful tool for qualitative assessments of survival probabilities, e.g. to gain first insight in effects of gear modifications within gears on survival probabilities.

The results of this study can contribute to more sustainable exploitation and conservation of skates in European waters. The relatively high survival probabilities may make skates eligible for exemptions to the landing obligation, effectively preventing 100% mortality among skates destined to be landed instead of discarded. Exemption from the landing obligation could favour population recovery [20]. The availability of representative discards survival estimates may benefit skate stock assessments as it provides better insight in the contribution of discarding to fishery mortality [17].

## Conclusions

We consider the discards survival probabilities that we report in this study for thornback and spotted skate as maximum values that are representative for year-round beam trawl, pulse trawl and flyshoot fisheries. Discard survival of thornback and spotted skate is affected by gear, water temperature, sea state, catch processing time and the condition in which skates are landed on deck. Vitality class provides a useful qualitative prediction of survival probability within gear. To further increase survival probability, it is recommended to keep catch processing time as short as possible and to dedicate research on gear modifications that reduce stressors inflicted upon fish during the catch and hauling process to increase the proportion of skates that is landed on deck in good condition. Additional data collection on survival probabilities under variable environmental and operational conditions will increase our understanding of the interactive effects of these conditions and support further decrease of mortality among skate discards.

## Supporting information

**S1 Table. Gear specifics of the pulse trawlers used for the discards survival trips.**
(DOCX)

**S2 Table. Gear specifics of the beam trawler used for the survival trips.**
(DOCX)

**S3 Table. Gear specifics of the flyshooter used for the survival trips.**
(DOCX)

**S4 Table. Data file.**
(CSV)

## Acknowledgments

The following contributors were indispensable for our research into discards survival of skates:

Owners, skippers and crews of all the participating fishing vessels for welcoming researchers with all their equipment on board and assisting them whenever needed; Mulder Transport BV for the transport of survival units from the fishing vessels to our laboratory in Yerseke; Visserijinnovatie Centrum Zuidwest Nederland for transporting all equipment to the fishing vessels and for maintaining all equipment in proper condition. PO Urk and Nederlandse Vissersbond for acquisition of the participating vessels. We thank the three anonymous reviewers for their valuable comments on the manuscript.

## Author Contributions

**Conceptualization:** Edward Schram, Pieke Molenaar, Jurgen Batsleer.

**Data curation:** Edward Schram, Lennert van de Pol, Katinka Bleeker, Jan Jaap Poos, Karolina Molla Gazi.

**Formal analysis:** Edward Schram, Lennert van de Pol, Jan Jaap Poos, Karolina Molla Gazi.

**Funding acquisition:** Edward Schram, Jurgen Batsleer.

**Investigation:** Edward Schram, Lennert van de Pol, Katinka Bleeker, Pieke Molenaar, Allard van Mens, Suzanne Cornelisse, Pim van Dalen, Wouter Suykerbuyk, Jurgen Batsleer.

**Methodology:** Edward Schram, Pieke Molenaar.

**Project administration:** Edward Schram.

**Supervision:** Edward Schram.

**Writing – original draft:** Edward Schram, Pieke Molenaar, Jurgen Batsleer.

**Writing – review & editing:** Edward Schram, Lennert van de Pol, Katinka Bleeker, Jan Jaap Poos, Jurgen Batsleer.

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
