## [Decision Letter · Decision Letter 0]

14 Jun 2024

PONE-D-24-20479Survival probabilities of thornback and spotted rays discarded by tickler chain beam trawl, pulse beam trawl and flyshoot fisheriesPLOS ONE

Dear Dr.  Schram,

Thank you for submitting your manuscript to PLOS ONE. After careful consideration, we feel that it has merit but does not fully meet PLOS ONE’s publication criteria as it currently stands. Therefore, we invite you to submit a revised version of the manuscript that addresses the points raised during the review process.

It is an interesting research dealing with an important field for the conservation of the entire marine ecosystem. Indeed, elasmobranchs are fundamental species for the well-being of marine ecosystems and the maintenance of ecological dynamics. The impact of the commercial fishery is one of the major concerns which heavily affect elasmobranch populations worldwide, also allowing the regional extinction of several species, as in the case of the Mediterranean Sea. 

We look forward to receiving your revised manuscript.

Kind regards,

Claudio D'Iglio, Ph.D.

Academic Editor

PLOS ONE

Journal Requirements:

  "European Union, European Maritime and Fisheries Fund (EMFF)"

5. Please provide a complete Data Availability Statement in the submission form, ensuring you include all necessary access information or a reason for why you are unable to make your data freely accessible. If your research concerns only data provided within your submission, please write "All data are in the manuscript and/or supporting information files" as your Data Availability Statement.

Additional Editor Comments:

Dear Doctor Schram,

The present Manuscript is well written, with a clear and reliable methodology and interesting results which strongly support the author's conclusion. There are some concerns to be fixed regarding the clarity of some parts of the Materials and Methods section, and some improvements which should be applied to the Discussion and Introduction sections to enhance their clarity and exhaustiveness.

Here are the reviewers' reports, please be careful to improve the Manuscript according to their suggestions.

All the bests,

Claudio D'Iglio, PhD

PLOS ONE

Accademic editor

Reviewers' comments:

Reviewer's Responses to Questions

**Comments to the Author**

1. Is the manuscript technically sound, and do the data support the conclusions?

Reviewer #1: Partly

Reviewer #2: Yes

Reviewer #3: Yes

2. Has the statistical analysis been performed appropriately and rigorously? 

Reviewer #1: Yes

Reviewer #2: N/A

Reviewer #3: Yes

3. Have the authors made all data underlying the findings in their manuscript fully available?

Reviewer #1: Yes

Reviewer #2: Yes

Reviewer #3: Yes

4. Is the manuscript presented in an intelligible fashion and written in standard English?

Reviewer #1: No

Reviewer #2: Yes

Reviewer #3: Yes

5. Review Comments to the Author

Reviewer #1: Dear Authors,

Thank you for your submission on the discard survival probabilities of thornback (Raja clavata) and spotted ray (Raja montagui). Your study provides valuable insights into this critical area of fisheries management. However, there are several areas where the manuscript would benefit from substantial revisions to improve its clarity and impact. Here are my detailed comments:

Introduction and Background:

Your introduction needs to provide a broader context for the study. The concept of discard survival should be framed within the larger scope of fisheries management and conservation. Additionally, it is crucial to cite foundational works that enhance the understanding of discard survival and the factors influencing it. In particular, the studies by Tiralongo et al. (2018) and Tiralongo et al. (2020) are fundamental for comprehending biological discards and survival, emphasizing that survival can sometimes be species-specific, as seen with Dasyatis pastinaca. Integrating these references will significantly strengthen your background section. See the attached pdf for further details.

Discussion:

The discussion needs to integrate your findings with existing literature more robustly. Again, referencing Tiralongo et al. (2018, 2020) will provide a deeper understanding of how species-specific traits influence discard survival. Discuss the implications of gear type on survival probabilities in greater detail and compare your findings with other studies on different species or in different regions. Provide a more comprehensive analysis of the interaction between water temperature, wave height, and survival probabilities. Explore why these factors might influence survival and how they interact. This analysis will add depth to your discussion and provide valuable insights.

Conclusion:

Your conclusion should be strengthened by explicitly stating the practical applications of your findings for fisheries management. How can these results be used to improve sustainability practices? Additionally, suggest areas for future research, particularly those that could address any limitations identified in your study or expand on the interaction effects observed.

Reviewer #2: The work "Survival probabilities of thornback and spotted rays discarded by tickler chain beam

trawl, pulse beam trawl and flyshoot fisheries" is well written and structured, easy to understand and deserves to be published after small corrections in the text

Reviewer #3: The manuscript titled Survival probabilities of thornback and spotted rays discarded by tickler chain beam trawl, pulse beam trawl and flyshoot fisheries reports on the discard survival of two ray species in 3 trawl fisheries in the North Sea. The methods and well-laid out and appropriate. The research was well-designed, although there was some variability in methods (number of control fish, number of rays per haul, timing in a haul that rays were collected), but not outside of what is to be expected for a field study. A really neat and tidy manuscript. The discussion covers a broad array of topics that are logical from their results.

The only think I’m not clear on is how discard mortality was recorded (see line 115 comment). Was each skate that came through the catch categorized as dead or alive, or just the ones sampled for the survival study? If the latter is the case, the dead rays were included in the tank study, but not actually placed in a tank?

Line-specific comments

Line 52: What measurement for minimum size? Total length? Disc width?

Line 60: Has any of these ray species been a choke species in a year? It’s good if they haven’t yet, but it would be good context to know how close these species have been to closing the fishery.

Line 69: What was the survival rate of Van Bogaert et al?

Line 115: Was data recorded (alive or dead) for all rays that came onboard? I think it’s also worth a statement in this section that condition of the ray was not a factor in choosing it as a test-fish, except if the ray was already dead (no spiracle movement for 15 seconds).

Line 127: Why did you focus on the big ones? Isn’t the minimum size 45-55 cm? Why 65 cm?

Line 130: I recommend moving this sentence to Line 126.

Line 266: Wasn’t the maximum time in a tank 21 days? I’m hesitant to put too much value in a leveling out of mortality at 20 days of a 21 day experiment. Looking at Fig 1a and 1c, I wouldn’t say the mortality rate has leveled off. This doesn’t change your results, just what you say about them.

Line 268: I’m not sure how direct mortality was recorded. Was every ray examined for direct mortality before discard? Or was this only of sampled rays? If the latter was the case, did you sample another ray from that haul to get to target numbers? See Line 115 comment also.

6. PLOS authors have the option to publish the peer review history of their article (what does this mean?). If published, this will include your full peer review and any attached files.

Reviewer #1: No

Reviewer #2: No

Reviewer #3: **Yes: **Kelsey C James

---

## [Author Response · Author response to Decision Letter 0]

27 Sep 2024

Please refer to the attached document for reviewer and editor comments.

---

## [Decision Letter · Decision Letter 1]

16 Oct 2024

PONE-D-24-20479R1

Survival probabilities of thornback skate (Raja clavata) and spotted skate (Raja montagui) discarded by tickler chain beam trawl, pulse trawl and flyshoot fisheries

PLOS ONE

Dear Dr. Schram,

Thank you for submitting your manuscript to PLOS ONE. After careful consideration, we feel that it has merit but does not fully meet PLOS ONE’s publication criteria as it currently stands. Therefore, we invite you to submit a revised version of the manuscript that addresses the points raised during the review process.

The Ms has been strongly improved accotding to the reviewers suggestion. There are only some minor refuses and modifications that should be improved and adjusted, as stated by reviewers in their comments.

We look forward to receiving your revised manuscript.

Kind regards,

Claudio D'Iglio, Ph.D.

Academic Editor

PLOS ONE

Journal Requirements:

Reviewers' comments:

Reviewer's Responses to Questions

**Comments to the Author**

1. If the authors have adequately addressed your comments raised in a previous round of review and you feel that this manuscript is now acceptable for publication, you may indicate that here to bypass the “Comments to the Author” section, enter your conflict of interest statement in the “Confidential to Editor” section, and submit your "Accept" recommendation.

Reviewer #1: (No Response)

Reviewer #2: (No Response)

Reviewer #3: All comments have been addressed

2. Is the manuscript technically sound, and do the data support the conclusions?

Reviewer #1: Yes

Reviewer #2: Yes

Reviewer #3: Yes

3. Has the statistical analysis been performed appropriately and rigorously? 

Reviewer #1: Yes

Reviewer #2: Yes

Reviewer #3: Yes

4. Have the authors made all data underlying the findings in their manuscript fully available?

Reviewer #1: Yes

Reviewer #2: Yes

Reviewer #3: Yes

5. Is the manuscript presented in an intelligible fashion and written in standard English?

Reviewer #1: Yes

Reviewer #2: Yes

Reviewer #3: Yes

6. Review Comments to the Author

Reviewer #1: The manuscript presents interesting and well-structured findings on the discard survival probabilities of Raja clavata and Raja montagui in different fishing activities. However, some major revisions are necessary to fully implement the work before potential publication. In particular, attention should be given to the methodology used for survival probability analysis, the presentation of results, and certain aspects of data interpretation. The bibliography and conclusions also require particular attention to ensure they accurately reflect the current state of research and draw stronger connections between the findings and broader implications for fisheries management. Specific comments are provided in the attached PDF for a thorough revision of the manuscript.

Reviewer #2: The MS in question is well structured and easy to read. However, the authors must explain in detail what is meant by "ray". The species considered are skates, so it is suggested to check the entire text and verify when it is necessary to use skates instead of rays.

Reviewer #3: The manuscript addressed all of the reviewer comments and is generally improved. The Introduction has a broader context that appeals to a wider audience. The methods and results are clearer. The discussion does a good job of contextualizing the results with other work. I only have a few line-specific comments to address below.

Line-specific Comments:

Line 49-51: You state that starry and common skates are discarded. Is this still true after the landing obligation came into effect in 2019. Do these two species have exemptions for the landing obligation? This is not clear in this paragraph. I recommend adding the species that currently have exemptions from the landing obligation at the end of the paragraph (Line 57) or the next paragraph (Line 64).

Line 78: In the abstract you include that survival probabilities are “affected by gear, catch processing time, wave height, and the interaction between water temperature and wave height.” I think it’s worth including the that you investigated predictors of survival as an objective.

Line 120: remove the third ‘belt’ in this line since I think you mean “manually collected by crew members.”

Line 128: Why do you use a cutoff of 65 cm rather than 55 cm. I understand why you want large individuals, but why the 10 cm difference?

Line 138-140: Please rewrite this sentence to enhance clarity. I think the point of this sentence is that you were able to sample more skates when skates died while at sea and sometimes you were also able to put two skates in one tank.

7. PLOS authors have the option to publish the peer review history of their article (what does this mean?). If published, this will include your full peer review and any attached files.

Reviewer #1: No

Reviewer #2: **Yes: **Fabrizio Serena

Reviewer #3: No

---

## [Decision Letter · Decision Letter 2]

5 Nov 2024

Survival probabilities of thornback skate (Raja clavata) and spotted skate (Raja montagui) discarded by tickler chain beam trawl, pulse trawl and flyshoot fisheries

PONE-D-24-20479R2

Dear Dr.Schram,

We’re pleased to inform you that your manuscript has been judged scientifically suitable for publication and will be formally accepted for publication once it meets all outstanding technical requirements.

Kind regards,

Claudio D'Iglio, Ph.D.

Academic Editor

PLOS ONE

Reviewers' comments:

Reviewer's Responses to Questions

**Comments to the Author**

1. If the authors have adequately addressed your comments raised in a previous round of review and you feel that this manuscript is now acceptable for publication, you may indicate that here to bypass the “Comments to the Author” section, enter your conflict of interest statement in the “Confidential to Editor” section, and submit your "Accept" recommendation.

Reviewer #1: All comments have been addressed

Reviewer #3: All comments have been addressed

2. Is the manuscript technically sound, and do the data support the conclusions?

Reviewer #1: Yes

Reviewer #3: Yes

3. Has the statistical analysis been performed appropriately and rigorously? 

Reviewer #1: Yes

Reviewer #3: Yes

4. Have the authors made all data underlying the findings in their manuscript fully available?

Reviewer #1: Yes

Reviewer #3: Yes

5. Is the manuscript presented in an intelligible fashion and written in standard English?

Reviewer #1: Yes

Reviewer #3: Yes

6. Review Comments to the Author

Reviewer #1: Dear Authors,

the manuscript can be now accepted in its current form. Well done and congratulations for this interesting work.

Reviewer #3: (No Response)

7. PLOS authors have the option to publish the peer review history of their article (what does this mean?). If published, this will include your full peer review and any attached files.

Reviewer #1: **Yes: **Francesco Tiralongo

Reviewer #3: No

---

## [Editor Report · Acceptance letter]

9 Dec 2024

PONE-D-24-20479R2 

PLOS ONE

Dear Dr. Schram, 

I'm pleased to inform you that your manuscript has been deemed suitable for publication in PLOS ONE. Congratulations! Your manuscript is now being handed over to our production team.

Kind regards, 

on behalf of

Dr. Claudio D'Iglio 

Academic Editor

PLOS ONE